# Long-term experimental evolution of HIV-1 reveals effects of environment and mutational history

**Eva Bons**[1], **Christine Leemann**[2,3], **Karin J. Metzner**[2,3]*, **Roland R. Regoes**[1]*

**1** Department of Environmental Systems Sciences, Institute of Integrative Biology, ETH Zurich, Zurich, Switzerland, **2** Division of Infectious Diseases and Hospital Epidemiology, University Hospital Zurich, Zurich, Switzerland, **3** Institute of Medical Virology, University of Zurich, Zurich, Switzerland

* karin.metzner@usz.ch (KJM); roland.regoes@env.ethz.ch (RRR)

## Abstract

An often-returning question for not only HIV-1, but also other organisms, is how predictable evolutionary paths are. The environment, mutational history, and random processes can all impact the exact evolutionary paths, but to which extent these factors contribute to the evolutionary dynamics of a particular system is an open question. Especially in a virus like HIV-1, with a large mutation rate and large population sizes, evolution is expected to be highly predictable if the impact of environment and history is low, and evolution is not neutral. We investigated the effect of environment and mutational history by analyzing sequences from a long-term evolution experiment, in which HIV-1 was passaged on 2 different cell types in 8 independent evolutionary lines and 8 derived lines, 4 of which involved a switch of the environment. The experiments lasted for 240–300 passages, corresponding to approximately 400–600 generations or almost 3 years. The sequences show signs of extensive parallel evolution—the majority of mutations that are shared between independent lines appear in both cell types, but we also find that both environment and mutational history significantly impact the evolutionary paths. We conclude that HIV-1 evolution is robust to small changes in the environment, similar to a transmission event in the absence of an immune response or drug pressure. We also find that the fitness landscape of HIV-1 is largely smooth, although we find some evidence for both positive and negative epistatic interactions between mutations.

## Introduction

Human immunodeficiency virus (HIV) is often considered a master evolver. The virus is constantly undergoing environmental changes, both at transmission—where it needs to adapt to a new host (reviewed in [1]) and within the host, where it colonizes various compartments and has to constantly adapt to changing immune responses [2–5]. Due to its high mutation rate [6–9] and large population sizes (reviewed in [10]), it can quickly adapt to these changing selection pressures.

The predictability of evolution is a central question in evolutionary biology that has been asked in the context of a wide range of organisms [11–14], including HIV [15–17]. The

**Data Availability Statement:** The alligned Illumina sequencing data are deposited in the NCBI short read archive under the Bioproject accession PRJNA681155. The processed data used to generate figures and statistics in the test, the data

from growth rate assays, simulation code and data and analysis scripts are available on Zenodo: 10.5281/zenodo.3787242.

**Funding:** RRR gratefully acknowledges the financial support of the Swiss National Science Foundation (www.snf.ch, grant numbers 31003A_149769 and 31003A_179170). The funders had no role in study design, data collection and analysis, decision to publish, or preparation of the manuscript.

**Competing interests:** The authors have declared that no competing interests exist.

**Abbreviations:** APOBEC3G, apolipoprotein B mRNA editing enzymecatalytic polypeptide-like 3G; HIV, human immunodeficiency virus; MOI, multiplicity of infection.

question does address not only which mutations occur under certain conditions, but also the order in which they occur—the evolutionary path. The predictability of evolution is determined by the environment and the genetic predisposition and counteracted by random processes.

The main factor influencing the evolutionary path of an organism is the environment in which it grows. The fitness of new mutations is often dependent on the surroundings, so mutations that are selected in 1 environment might be selected against in another, causing patterns of mutation and reversion. The areas of the genome where these environment-specific mutations occur are interesting since they indicate areas of the genome that potentially code for important interactors with environmental factors. Environmentally robust mutations, on the other hand, are fit in a wide range of environments and ensure that an organism can survive in fluctuating surroundings. When many of the available mutations are environmentally robust, many mutations will have the same, or similar, effects across environments, which will lead to high levels of parallel (or convergent) evolution, where the same mutations are found in independent populations [18–21].

While the fitness of an organism is largely dependent on the environment, the genetic background of a new mutation can also play a major role. Mutations that have come up in the past can further impact evolution due to epistatic effects with new mutations. The relationship between genotype and fitness is often conceptualized as the fitness landscape (reviewed by [22,23]). The rougher the fitness landscape, or the more epistatic effects there are, the more the mutational history of an organism will matter for the future evolutionary path. In some cases, a mutation that in itself has no large fitness effect can completely change the path of evolution because of the interactions it has with other mutations. This concept of historical contingency is central in the study of history and evolution [24] and has been shown to be important for evolution various organisms, for example, in bacteria [22,25] and yeast [26,27].

Together, history and environment shape the evolutionary paths that are possible for an evolving population, but the exact paths taken will largely depend on random effects, such as mutation, drift, and linkage. In a population with a high mutational supply, such as HIV, the majority of mutations are expected to be present at any given time. Evolution will then behave more deterministically, with a large probability of fixing the largest effect mutation first. However, drift and linkage will still influence the route taken, as bottlenecks can increase drift and linkage of neutral mutations with beneficial mutations, which can be important in historical contingency.

Fig 1 shows some examples of how the impact of history and environment can imprint in the genome over the course of evolution.

Experimental evolution, in combination with sequencing of the evolved clones, is often used to study the evolution of an organism in detail (see [28] for an overview of notable experiments). Most of these experiments are performed with bacteria, but the unique evolutionary dynamics of viruses makes them an interesting subject for evolution studies as well. Bacteriophages are a favorite study organism [21,29,30], due to their fast replication cycles and large population sizes. For viruses with a human host, experimental evolution has been used to estimate the distribution of fitness effects in poliovirus [31] and the recovery potential of HIV after severe bottlenecking [32].

In this study, we present results from the longest running HIV evolution experiment to date. We infer the contributions of environment and history to the evolution of HIV, using the patterns in the time series of new mutations occurring over the course of an evolutionary experiment where HIV-1 is adapting to 2 different cell types. With enough time to evolve, and enough detail in the sequenced populations, we can study aspects of the fitness landscape without directly measuring the fitness of each mutation in various genetic backgrounds and environments.

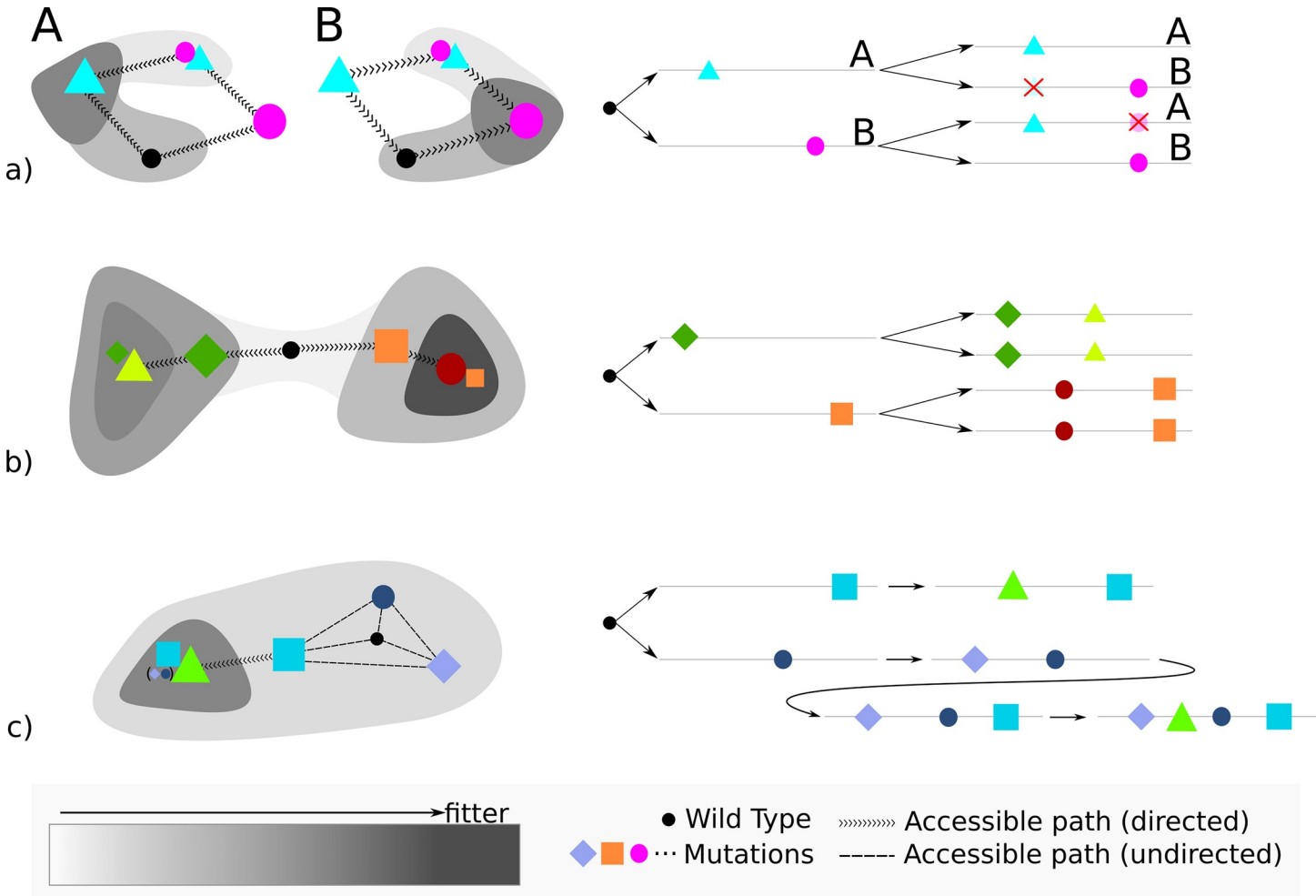

**Fig 1. Minimalistic examples of the effect of history and environment.** Left: simplified fitness landscapes, which only show accessible paths that increase fitness. Darker shading indicates increased fitness. Right: some possible realized paths from the wild type (black dot). Gray lines represent genomes; colored shapes indicate mutations. (**a**) Environmental effects on non-robust mutations. Two possible mutations: one is fit in environment A, but not in B, and the other is fit in environment B, but not in A. Upon environmental change, the unfit mutation will eventually revert to the wild type and the other beneficial mutation will fixate. (**b**) Historical effect—contingency: From the wild type, there are 2 options to increase fitness. While the first step will depend on random processes, the next step is defined by the history, since there is only 1 option to increase fitness from there. The fitness maximum for this example (orange/red) will not be reached if the first step taken is the green one. (**c**) Historical effect—gateway mutations: The fitness maximum (green triangle) is only accessible via a neutral mutation (large blue square) that has to happen first. Other neutral mutations might happen before the potentiating mutation is found and opens up the path toward the fitness optimum.

## Results

To determine the contributions of environment and history on HIV-1 evolution, the long-term evolution of the virus needs to be monitored in different environments. HIV-1 was therefore passaged on 2 different human T-cell cultures, MT-2 and MT-4, for 240 to 300 passages, in 8 (mostly) independent lines, and 8 derived lines (see Materials and methods for a more detailed overview of the lines), corresponding to approximately 400 to 600 generations. Four of the derived lines underwent a change in environment: They evolved on 1 cell type for 50 passages and on the other for 220 successive passages. This experiment constitutes the longest evolutionary experiment with HIV to date, lasting 2 to 3 years of evolution.

Results from a subset of these experiments were already presented in [20], highlighting the abundance of parallel evolution in the experiment. These high numbers of parallel mutations

are not compatible with neutral evolution, indicating the presence of strong evolutionary pressures, even though the virus is evolving in a very constant environment without adaptive immune or drug pressure.

## Evolution is non-neutral throughout the experiment

Despite the constant environment and low selection pressures, the virus is continuously evolving.

Over the course of all experimental lines, 1,520 mutations reach majority independently, and many of which are shared. In total, we see 823 unique mutations reaching majority. Since the HIV genome is 9,719 bases long, and every position has 3 possible mutations, there are 29,157 possible mutations. Hence, almost 3% of all possible mutations reach majority. A total of 41% of these mutations (378) are shared—they occur independently in more than 1 line—with 93 mutations occurring in more than 3 lines, and 2 mutations occurring in all 16 lines. Over half of these majority mutations (415), reach fixation (a frequency of 0.99 or higher) in any of the lines.

Using simulations of neutral evolution, we estimated that when the mutation rate is $10^{-5}$ [6] around 975 mutations are expected to reach majority, of which 15 are shared. Only 12 mutations are expected to reach fixation. If the mutation rate is higher, as suggested in [9], at $10^{-4}$, we see 10 times more unique majority mutations than in the data (9,500); however, sharing of mutations is much more limited than in the data: Only 20% of mutations is shared, of which none occurred in more than 6 lines. Even here, only 327 mutations reach fixation, still less than what we see in the data. The discrepancy in the numbers of fixed, shared, and majority mutations between the data and neutral simulations indicates that evolution is non-neutral in the HIV long-term evolution experiment.

While there is a difference in apolipoprotein B mRNA editing enzyme, catalytic polypeptide-like 3G (APOBEC3G) expression between the 2 cell types, HIV has a fully functional *vif* gene, so that we do not see any difference in G-to-A mutation rates between the lines, suggesting that APOBEC-mediated mutations are not relevant in this dataset (see S2 Text).

Growth rates were measured for the ancestor HIV-1 NL4-3 and 4 time points of 4 out of 16 evolutionary lines. The results from this experiment are consistent with non-neutral evolution: Later time points have higher growth rates in the line they were passaged in than earlier time points (Figs A and B in S3 Text). However, the effect is not restricted to the cell type the virus was passaged on. Some of the adapted viral populations also perform better in the other cell type, indicating some level of transferability of adaptation.

## Reversion mutation reach higher frequencies than other mutations

The majority mutations that occur throughout the experiment are distributed relatively equally across the genome, with some highly shared mutations being silent, or in noncoding regions. For most of the mutations that reach majority in this dataset, we cannot determine their exact function from the existing literature. Additionally, even though some mutations have been characterized before, our findings will be biased toward drug-related and immune escape-related mutations, since they are the mutations that are most often characterized.

One type of mutation that we can identify reliably and without bias are reversions, mutations back to the overall HIV consensus sequence (the so-called consensus of consensus) at positions where the ancestral virus differs from this consensus HIV sequence. There are 831 of these potential reversion sites in the ancestral virus, at 76 of which, a reversion mutation reached majority in at least one of the evolutionary lines. This means that 9% of the unique mutations we observe are reversion mutations.

Overall, reversions tend to reach higher frequencies than other mutations, even compared with non-reversion mutations at reversion sites (the average maximum frequency a mutation reaches is 14% for reversions, 6% for non-reversion mutations at reversion sites, and 7% for other mutations. A permutation test results in $p < 0.01$ for all comparisons). While they reach higher frequencies, reversion mutations are not more shared than other mutations.

## Higher mutational supply but also lower fixation rates in MT-4

In all evolutionary lines, mutations accumulate rapidly, as seen in Fig 2. However, we find a striking difference in the pattern at which mutations accumulate. In virus passaged on MT-4 cells, majority mutations accumulate linearly at a rate of 2 mutations per 10 passages. In MT-2, the virus accumulates majority mutations twice as fast, although the rate is not constant over time.

Minority mutations—present at a fraction of at least 0.05—increase similarly in both environments in the first 100 passages. After that, the number of minority mutations stops increasing in MT-2, while virus evolving on MT-4 cells keeps accumulating minority mutations at a similar rate as in the first 100 passages. This saturation in minority mutations in MT-2 is likely due to selective sweeps removing low-frequency mutations.

Four of the evolving lines underwent an environmental switch, from MT-2 to MT-4 and vice versa, at passage 50. While the accumulation of minority mutations immediately follows the trend of the new environment in these switched lines, there appears to be a "memory" of the initial environment in the accumulation of majority mutations, with patterns of accumulation somewhere in between the lines that did not undergo a switch.

The number of mutations at passage 240 (the last passage reached by all experimental lines) is significantly impacted by both the initial and final environments for both minority and majority mutations ($p < 0.05$ for all groups, 2-way ANOVA), highlighting the lasting imprint of the initial environment on mutation accumulation.

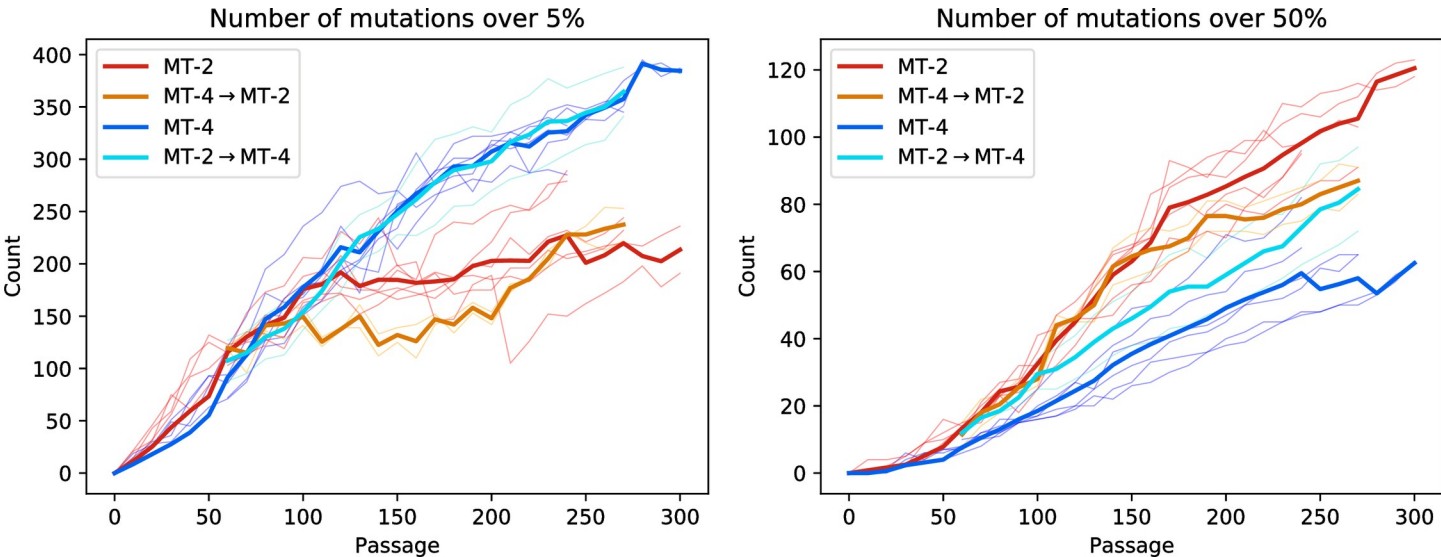

**Fig 2. Accumulation of minority mutations (left) and majority mutations (right) relative to the ancestral sequence in the different experiments.** The thick lines show the mean number of mutations above the threshold for all the different experiments, and the thin lines show the individual counts for each evolutionary line. In total, there are 16 lines, 6 per single environment (MT-2 and MT-4), and 2 per switch (MT-2 → MT-4 and MT-4 → MT-2). The raw data underlying this figure and associated analysis code are available at https://doi.org/10.5281/zenodo.3787242.

One of the striking differences between the environments is the temporal dynamics of new mutations. Fig 3 reveals that mutations in virus evolving on MT-4 tend to slowly rise to fixation, with little interference between mutations. In MT-2, on the other hand, the time series of mutations are much more erratic. This, together with the saturation dynamics in minority mutations, suggests that there are stronger selective sweeps in virus evolving on MT-2 cells. This results in fewer minority mutations due to the sweep dynamics and simultaneously more majority mutations due to hitchhiking.

## Differing patterns of mutations between the environments suggest adaptation to environment

To investigate if the environmental effects go beyond the dynamical differences in mutation accumulation between the 2 environments, we tested whether there are environment-specific mutations, as illustrated in Fig 1A.

We find that over 40% of mutations that reach majority occur in more than 1 line, and the majority of which occur in both environments (Fig 4). Still, there are some shared mutations that occur in 1 environment only. Some of these mutations will occur in only 1 environment by chance—they simply have not occurred in the other environment yet—but some might be truly environment-specific mutations, positions where the fitness landscape differs between the 2 environments.

To test whether some of these mutations might be environment-specific, we used randomization tests with the assumption that mutations are equally likely to occur in each

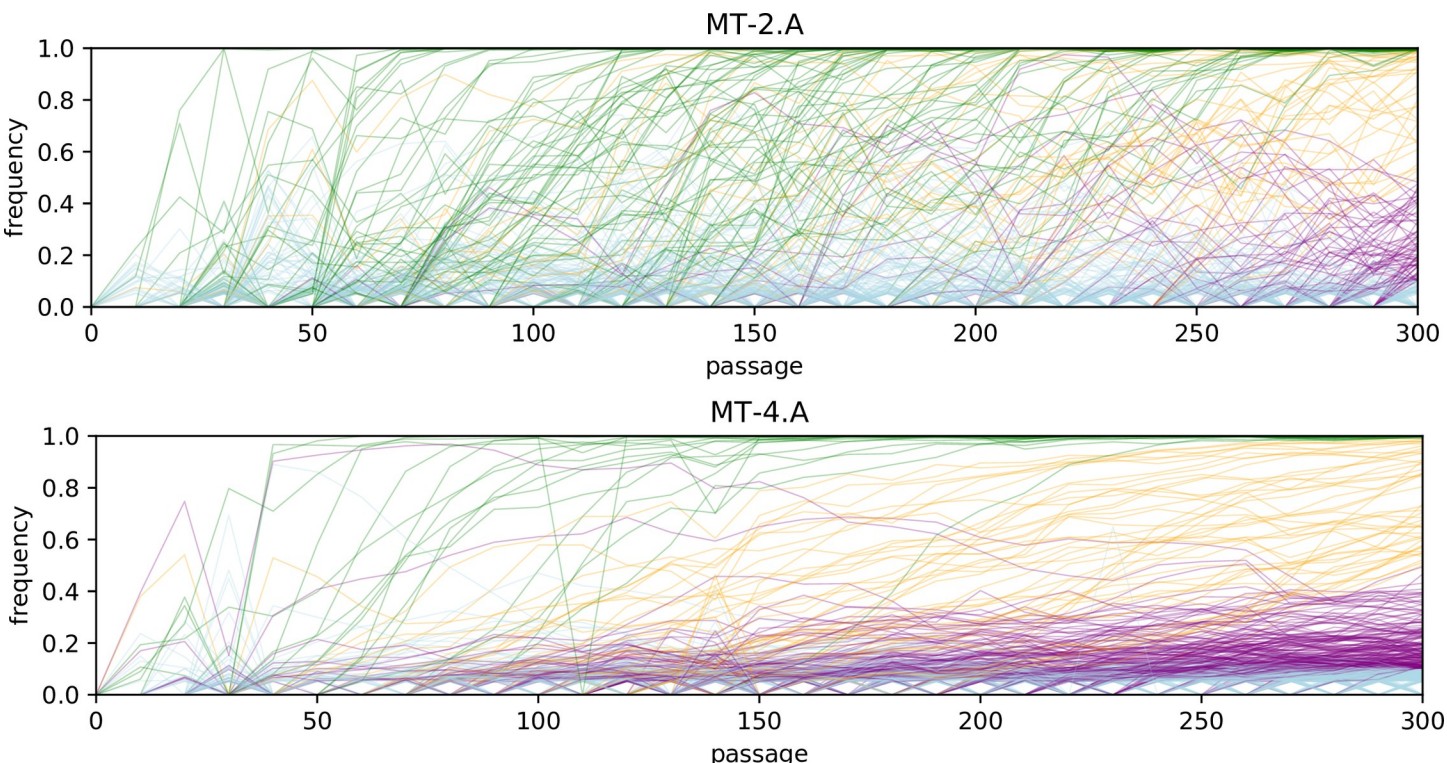

**Fig 3. Time series of all mutations occurring in 2 evolutionary lines: MT-2, replicate A (top) and MT-4, replicate A (bottom).** Every line is a mutation. Green lines indicate mutations that have reached fixation (99% or more) by the last passage, and orange lines indicate mutations that have reached majority by the last passage. Purple lines are still present as a minority (>5%) and blue lines are extinct (<5%) by the last passage. The raw data underlying this figure and associated analysis code are available on https://doi.org/10.5281/zenodo.3787242.

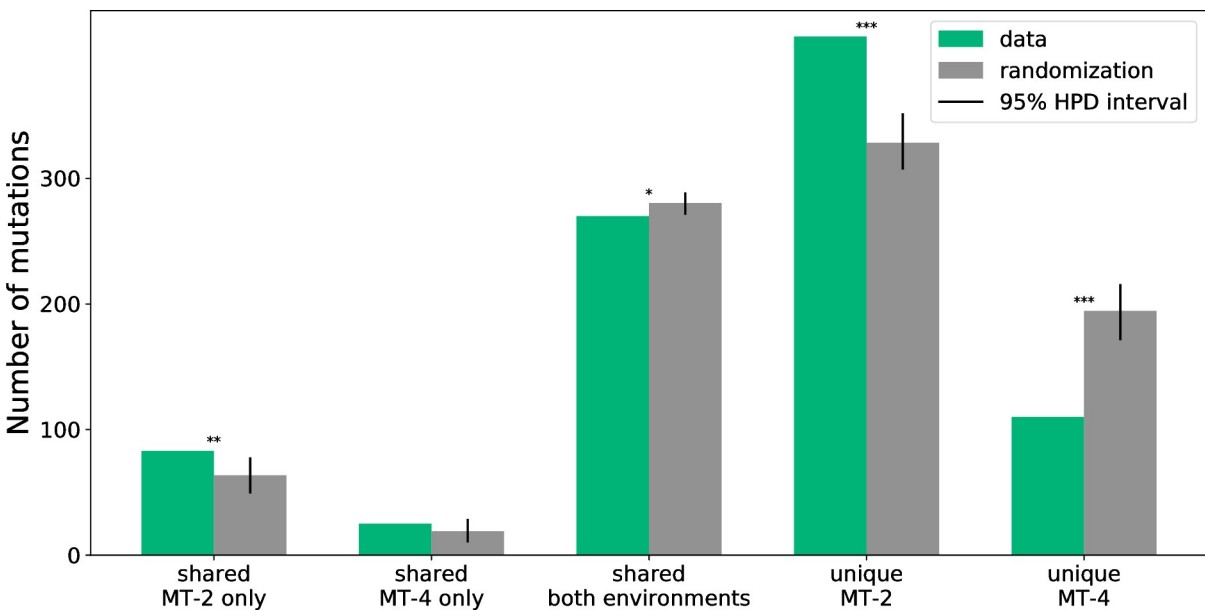

**Fig 4. Classification of all mutations in the dataset (green) compared with the expectation from randomization (gray, mean of 1,000 randomizations).** Black lines indicate the 95% highest probability density interval of the randomization. Shared mutations are mutations that reach majority in more than 1 evolutionary independent line and are only counted once (the same mutated position in 2 lines counts as 1 mutation). *p*-Values are indicated as follows: *p < 0.05, **p < 0.01, ***p < 0.001. The raw data underlying this figure and associated analysis code are available at https://doi.org/10.5281/zenodo.3787242.

environment. Fig 4 shows the results of these randomization tests: Under the null assumption of no environmental impact on sharing, one would expect even more between-group sharing and less environment-specific sharing than currently observed. However, the effect is small, with only approximately 25 mutations that are expected to be environment specific out of 108 mutations that are shared in only 1 environment. There are also many more unique mutations (mutations that only occur in a single evolutionary line) present in MT-2—and many less in MT-4—than expected. This means that mutations occurring in MT-4 are more likely to be shared than in MT-2.

The high amount of unique mutations in MT-2 could be due to hitchhikers—mutations that are not necessarily beneficial themselves but that occur in the background of another beneficial mutation—which rise along due to linkage. This is in line with the observations in the section on "Higher mutational supply but also lower fixation rates in MT-4".

## The genetic background matters more for MT-4 than for MT-2

Besides environmental effects, the genetic background might also be important as to which mutations reach majority. Interactions between mutations can lead to patterns of co-occurrence and exclusivity (either observing 1 mutation or the other, but never both) of mutations that inform us about the importance of the historical mutations and the shape of the fitness landscape. Historical contingency and epistatic effects are typically studied by looking into the order of occurrence or linkage of mutations. The data presented here does not allow us to look into the order of mutations in enough detail, because many mutations reach majority at the same time. Due to the short read population-level sequencing approach, linkage information is also not available. We have therefore developed statistical techniques based on the frequency of co-occurring mutations to understand the effect of epistatic interactions on the evolutionary paths the virus takes in this experiment.

We can, for example, look at always-combinations—mutations that do not occur unless another mutation occurred before. This could be an indication of positive epistasis between the mutations, as illustrated in Fig 1C. We can only look at always-combinations if a mutation reached majority in at least 2 independent lines. It can be expected that some mutations always occur in the background of another, just because it is a mutation that tends to occur early or by pure chance. In other words, always-combinations can arise for other reasons that epistatic interactions. A similar logic exists for never-combinations, mutations that are never observed together with another mutation, indicating negative epistasis. This is in line with Fig 1B.

To determine the extent of epistatic interactions from always- and never-combinations, we developed a randomization procedure that predicts the frequency of these combinations assuming no epistatic interactions. Our randomization procedure adjusts for differences in the time of first emergence of mutations, which could affect the expectations.

On average, shared mutations have 4 to 6 mutations that are always in the background. In virus evolving on MT-2, we see no more always-combination than expected if there are no interactions between mutations. In MT-4, the virus tends to have 3 to 4 more mutations that are always in the background than expected (Fig 5). This difference is highly significant ($p$-value of $6.81 \times 10^{-8}$, Wilcoxon signed-rank test). We also see several mutations that have over 5 extra mutations always in the background, an indication for higher-level interactions.

In both lines, there are many more never-combinations than expected. Again, the effect is larger in MT-4, where we see 26% more never-combinations than expected ($p$-value of $6.58 \times 10^{-31}$, Wilcoxon signed-rank test). In MT-2, we see 20% more never-combinations ($p$-value $1.22 \times 10^{-58}$, Wilcoxon signed-rank test).

## Discussion

In this study, we investigated the effect of environment and history on the evolution of HIV-1 in a long-term evolution experiment, where the virus evolves in a stable environment in the absence of any strong selection pressures such as drugs or immune pressure. In order to minimize the effects of selection, we chose the HIV-1 strain NL4-3. Even though this strain is quite different from clinical isolates, it is well adapted to lab conditions and replicates well in the 2 environments (MT-2 and MT-4 cell lines) used in our experiment. The long-term nature of the experiment allows for the natural accumulation of many mutations, allowing us to understand the evolutionary dynamics of HIV evolution using novel statistical approaches.

A large amount of mutations that reach majority in any of the evolutionary lines is shared, in most cases even across environments. This means that the majority of beneficial mutations are advantageous across the 2 cell environments and that there are only few environment specific mutations. About 9% of mutations that reach majority can be classified as reversions to the consensus. These mutations were likely adaptations to a previous environment that now are no longer beneficial. We also find a handful of highly shared mutations that are synonymous or located in untranslated regions. This indicates that HIV evolution is not only taking place at the protein level. Highly shared mutations in the 5′ untranslated region are likely related to viral packaging and regulation [33], while highly shared synonymous mutations could indicate codon optimization or selection for certain RNA structures, as shown by Zanini and colleagues [34].

We also find evidence that some mutations are only fit in certain mutational backgrounds, but only for virus evolving on MT-4 cells. This indicates that there are some positive epistatic interactions between mutations in MT-4. However, this does not exclude the presence of positive epistasis in MT-2, as we can only detect interactions that involve mutations that are lethal or deleterious on their own. In both lines, we also find evidence for incompatibility between some mutations, an indication of negative epistasis.

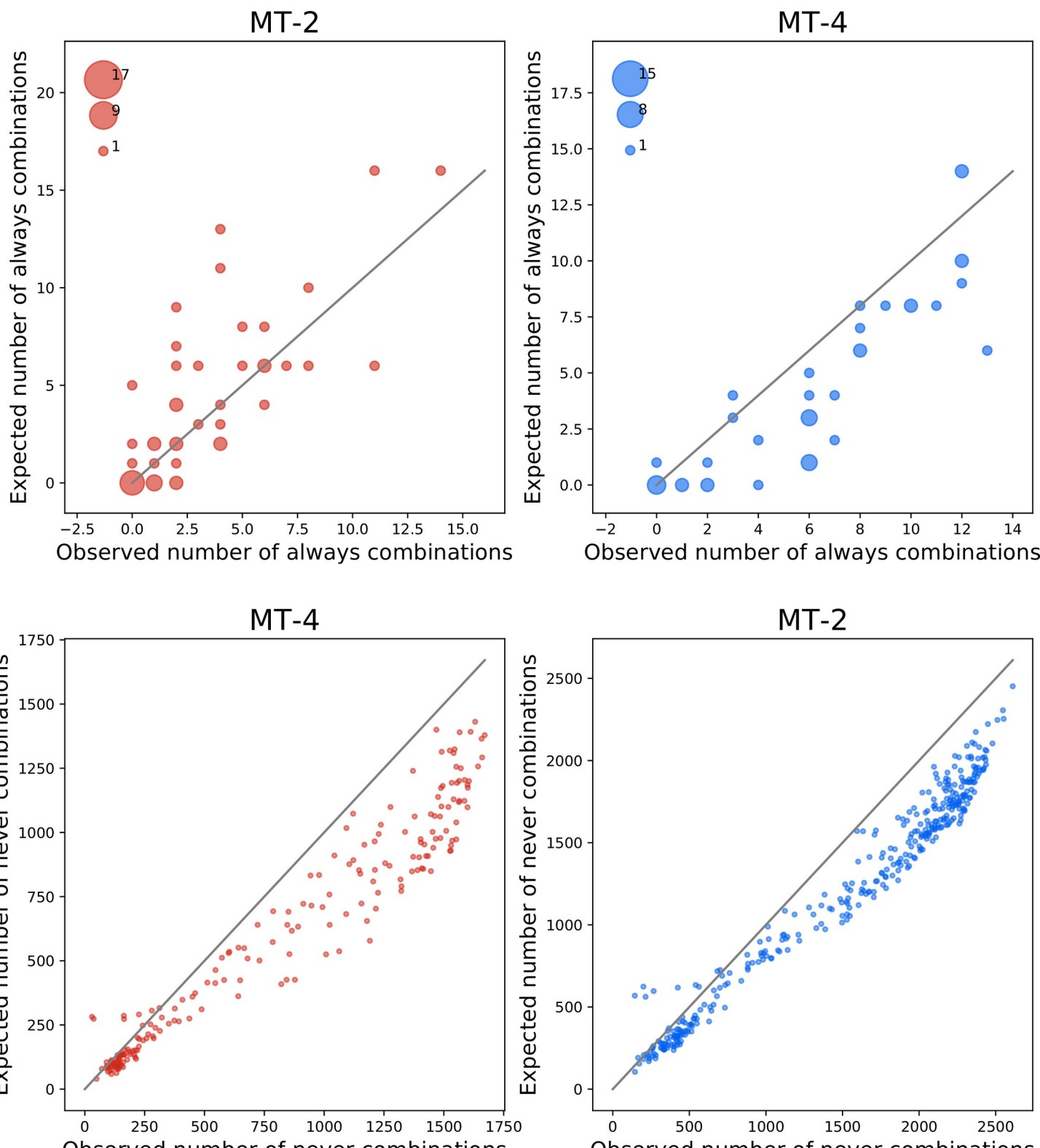

**Fig 5. Comparison of the number of always-combinations (top) and never-combinations (bottom) in the data to randomizations.** Gray lines indicate equality. There are significantly more always-combinations than expected in MT-4 cells and more never-combinations than expected in both environments. The raw data underlying this figure and associated analysis code are available at https://doi.org/10.5281/zenodo.3787242.

While epistatic interactions in HIV-1 have been demonstrated before [35–37], we find evidence for interactions between more than 2 mutations and a potential impact of the environment. Most mutations, however, seem to not interact and the same mutations tend to reach majority in the different evolutionary lines with different mutational backgrounds. This leads to the conclusion that the fitness landscape is mostly smooth, with many paths toward the same fitness peak.

The environment, in this case 2 different T-cell lines on which the virus evolves, has a large impact on the patterns of mutation accumulation. While the differences between the environments are small (they are the same cell type, both human T-cell lymphotropic virus type 1 – positive, derived from different individuals), the mutational patterns between the environments are remarkably different: In 1 environment (MT-2), majority mutations accumulate twice as fast as in the other. It appears as if there are many strong selective sweeps in MT-2, while the virus evolving on MT-4 steadily accumulates mutations without strong sweeps.

These differences in evolutionary dynamics could be responsible for the large discrepancy in the number of unique majority mutations in the 2 environments. In MT-2, the strong selective sweeps take along some hitchhikers, which are likely not selectively advantageous and are therefore unlikely to be shared. In fact, direct measures of growth rates show that the lines evolving on MT-2, with suspected hitchhikers, perform less good on MT-2 cells than virus that evolved on MT-4, with fewer suspected hitchhikers.

Besides differences in fitness effects, the differences in mutation accumulation could be due to differences in transfer sizes, since for most of the experiments, the transfer size for MT-4 lines was double that of the MT-2 lines. However, the pattern of differing mutation accumulation is still present (although decreased) between lines where transfer sizes were equal. Another possible explanation for the differences in mutation accumulation between the lines is a difference in mutation rate. The initially very similar increase in minority mutations, however, indicates that the mutation rates do not differ substantially between the 2 environments. While this rate is not independent of selection—a mutation cannot have a very negative effect if it can reach a fraction of 0.05 in a population—it is less sensitive to differences in the fitness landscape and indicates that there is a similar supply of nonlethal mutations in both lines. The later saturation of these minority mutations in MT-2 can be explained by the observed sweep dynamics in this line: The sweeps wipe out the diversity buildup faster than mutation can replenish them. It also explains the difference in the accumulation of majority mutations. Beneficial mutations take along hitchhikers, mutations that are not necessarily beneficial but are linked to a beneficial mutation, thereby increasing the amount of majority mutations.

The differences in evolutionary dynamics between the 2 environments are thus likely to be due, at least partly, to small differences in fitness landscape. The large amount of shared mutations between the 2 environments indicate that HIV-1 is quite environmentally robust, but the patterns of mutations indicate that evolution in MT-2 is less constrained than in MT-4. This might be due to the presence of more and higher effect beneficial mutations, which leads to stronger sweep dynamics and less interactions between mutations. This results in more neutral or low-effect mutations that can hitchhike along with higher effect mutations, and less evidence for historical contingency, shown by the absence of an excess of always-combinations.

While the dataset presented here allows us to look into the evolution of HIV-1 with respect to environment and history in great depth, there are some limitations to the experimental setup and execution. While the multiplicity of infection (MOI) was kept low over the course of the experiment, we cannot exclude recombination. The population-level sequencing also does not allow to look into the linkage of mutations, which could provide more explicit evidence for our hypothesis on selective sweeps and positive/negative epistasis. We also know of a contamination event early on between replicates A and B of both environments (see S1 Text). A change

in experimental protocol made contamination much less likely in the other experimental lines. However, it still makes some of the lines not completely independent. We therefore excluded these lines from some of our analyses.

This long-term evolution experiment was carried out under laboratory conditions in 2 highly constant environments—very different from the natural environment of HIV, which is constantly changing due to the immune response, migration between within-host compartments, and transmission to a new host. The tightly controlled environment can still give us great insight into why the virus is so adept at surviving these constant changes in the environment. Firstly, the largely smooth landscape prevents it from getting stuck in an unfavorable part of sequence space. Secondly, many adaptations observed in the experiment occurred in both environments, indicating strong environmental robustness. Considering that the cell cultures used were originally derived from 2 distinct individuals, we can consider the differences between environments in the experiment similar to the environmental changes at a transmission event. Additionally, the high mutation rate means that the adaptation can happen fast and reliably. Within a surprisingly short time, the same adaptive mutations are fixed in independent evolutionary lines.

Studies of experimental evolution are a common tool to understand the evolutionary dynamics of an organism. To study viral evolution, the bacteriophage φX174 has been used as a model organism, and many parallels can be drawn between our results and earlier work. Across evolutionary experiments with φX174, large amounts of parallel mutations are found [21,29,30,38]. Unlike our experiments, most of these studies involved strong selection pressures. An exception in this regard is [30] in which the virus was allowed to evolve in the absence of strong selection in a chemostat environment for 180 days—roughly 12,960 generations. Not only did they observe mutations that arose in previous evolution experiments with φX174, but they also found that the phage continued to adapt throughout the experiment. In addition, several of these studies identified adaptation at synonymous sites, indicating that selection is not only operating at the protein level.

In most evolutionary studies with viruses, levels of observed historical contingency are low. In the study by Wichman and colleagues [38], for example, φX174 evolved in a chemostat under strong selection pressure exerted by high temperatures and a new host species. While many of the observed mutations appeared in multiple evolutionary lines, the order of appearance was not conserved. Some level of historical contingency was observed in an evolutionary study with Tobacco Etch Virus [39], which was performed under specific selection pressures.

While parallel evolution is common and the role of historical contingency is low in most viral systems, the opposite appears to be true in cellular systems, such as *Escherichia coli* and *Saccharomyces cerevisiae*. For example, in Lenski's long-term evolution experiment, in which 12 populations of *E. coli* have been passaged in liquid culture for over 60,000 generations, only 5% to 10% of observed point mutations are shared between the independently evolving populations [40]. In our study, in contrast, over 40% of the majority mutations were shared. Moreover, in Lenski's long-term evolution experiment, there is evidence for strong historical contingency for at least 1 adaptation that confers the ability to use citrate [25]. In a smaller-scale experiment with the yeast *S. cerevisiae* [27], in which 3 populations were allowed to independently evolve under a serial passaging regime for 5 months, phenotypic convergent evolution, but no parallel mutations, were observed. This study also provides strong evidence for historical contingency.

The differences in evolutionary dynamics between viruses and cellular life can be explained—at least in part—by the differences in population genetic parameters. Viral mutation rates per site are typically many orders of magnitude larger than those of cellular systems. For comparable population sizes in evolution experiments with viruses or cellular organisms, the

higher viral mutation rates will lead to many more mutations per site, and hence to more parallelism. Beyond their explanation, the differing outcomes of evolutionary experiments with viruses and cellular organisms stress how important it is to study evolution across organisms in different population genetic regimes—an undertaking to which our study contributes.

Even among viruses, HIV-1 is a master evolver, not only because of its high mutation rates and large population sizes, but also because of its fitness landscape that is smooth and robust, allowing it to easily overcome environmental changes.

## Materials and methods

### Passaging of HIV-1

The human T-cell leukemia cell lines MT-2 and MT-4 [41] were obtained through the AIDS Research and Reference Reagent Program, Division of AIDS, NIAID, NIH from Dr. Douglas Richman. Cells were maintained in RPMI 1640 medium (Sigma-Aldrich, Buchs, Switzerland) containing 10% fetal calf serum, 100 U/ml penicillin, and 100 μg/ml streptomycin. The HIV-1 full-length plasmid pNL43 was obtained through the AIDS Research and Reference Reagent Program, Division of AIDS, NIAID, NIH from Dr. Malcolm Martin [42]. The virus stock HIV-1 NL4-3 was generated and characterized as previously described [43].

For experimental lines MT-2.A, MT-2.B, MT-4.A, and MT-4.B (Fig 6), the cultures were grown in 12 well plates, where the first row consisted of infected MT-2 cells and the third row of infected MT-4 cells. Uninfected control cells in the second row never showed signs of HIV-1 infection. At day 0, $2 \times 10^5$ cells per replicate and per cell line were infected with HIV-1 NL4-3 at an MOI (multiplicity of infection measured on peripheral blood mononuclear cells) of 0.0007 resulting in 4 independent cell cultures. We also measured the MOI at passage 90 of the experiment, which ranged between 0.012 and 0.064. Hence, coinfection should be relatively

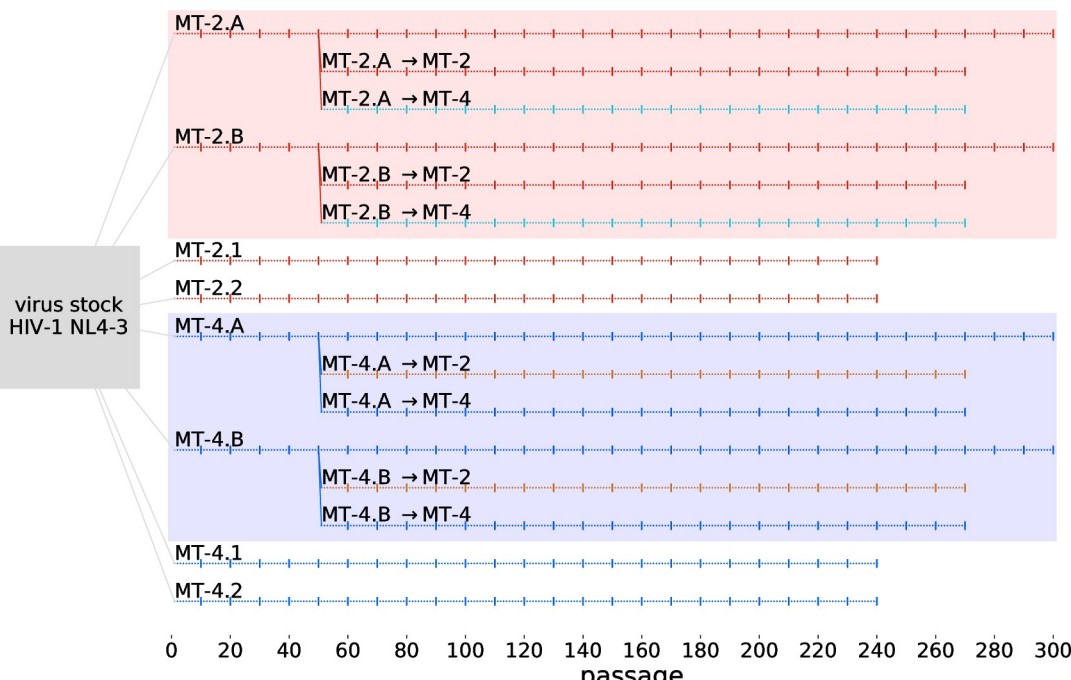

**Fig 6. Overview of the experimental design.** Shaded areas indicate lines that are not completely independent due to shared history or contamination events. Results from passage 10–90 of lines MT-2.1, MT-2.2, MT-4.1, and MT-4.2 have been published in [20].

rare in our experiment. Virus passaging was performed twice a week (alternating 3 and 4 days) as follows: Infected cell cultures were resuspended. A total of $2 \times 10^5$ uninfected MT-2 cells were inoculated with 1 μl cell suspension and MT-4 cell lines with 3 μl.

For the other experimental lines, the cultures were grown in separate cell culture flasks.

For experimental lines MT-2.1, MT-2.2, MT-4.1, and MT-4.2, at day 0, $4 \times 10^5$ cells per replicate and per cell line were infected with HIV-1 NL4-3 at an MOI of 0.01 resulting in 4 independent T-cell cultures. Virus passaging was again performed twice a week (alternating 3 and 4 days) as follows: Infected cell cultures were resuspended. A total of $4 \times 10^5$ uninfected cells were inoculated with 2 μl cell suspension. In the first 25 passages, higher volumes (30 to 3 μl) were transferred based on the extent of cytopathic effects microscopically observed in the cell cultures.

To start the derived experimental lines (MT-2.A→MT-2, MT-2.A→MT-4, MT-2.B→MT-2, MT-2.B→MT-4, MT-4.A→MT-2, MT-4.A→MT-4, MT-4.B→MT-2, and MT-4.B→MT-4), 20 μl (for MT-2 derived lines) or 60 μl (for MT-4 derived lines) was added to $4 \times 10^5$ cells per replicate. Virus passaging was again performed twice a week (alternating 3 and 4 days) as follows: Infected cell cultures were resuspended. A total of $4 \times 10^5$ uninfected cells were inoculated with 1.5 μl cell suspension for lines evolving on MT-2 and 4.5 μl cell suspension for lines evolving on MT-4.

Across all experiments, cell-free supernatant was stored at −80˚C every 10th transfer.

## Sequencing

Near full-length genomes of the virus stock HIV-1 NL4-3 (ancestor) and every 10th passage for all experiments were sequenced using the Illumina MiSeq next-generation sequencing platform (Illumina, San Diego, California, United States of America) as previously described [43]. Briefly, HIV-1 RNA was isolated from 150 μl virus stock HIV-1 NL4-3 or cell-free supernatant. Five overlapping amplicons were generated by reverse transcription polymerase chain reaction covering almost the full genome of HIV-1 per sample. The 5 amplicons per sample were pooled, and libraries were prepared with the Nextera XT DNA Sample Preparation Kit (Illumina) according to the manufacturers description. Next-generation sequencing was performed using a MiSeq Benchtop Sequencer (Illumina) generating paired-end reads of $2 \times 250$ bp length (v2 kit). To minimize the risk of cross contamination, samples from each replicate line were processed separately.

## Measurements of growth rates

Growth rates were measured for the wild type and for passages 60, 120, 180, and 240 of lines MT-2.1, MT-2.2, MT-4.1, and MT-4.2.

For this, $4 \times 10^5$ cells were plated on eight 6-well plates (4× MT-2 cells, 4× MT-4 cells). For growth rate measurements of the wild type, 2, 10, and 20 μl of the NL4-3 stock were added to a well containing MT-2 cells and to a well containing MT-4 cells. One MT-2 well and 1 MT-4 well were left without virus.

For the evolved lines, 2 μl of the cell-free supernatant was added to the wells containing MT-2 or MT-4 cells, resulting in 32 cultures (4 lines, 4 passages, and 2 cell types).

Plates were incubated for 4 days at 37˚C, 5% $CO_2$. At day 1, 2, 3, and 4, 50 μl was transferred to p24 ELISA plate and measured.

## Raw data processing

Raw reads were aligned to the reference (NL4-3, Genbank accession number AF324493.2) using BWA-MEM. Only reads with a mapping quality larger than 35 where considered for

further analysis. Mutations were then identified from the aligned reads. As an ancestor, the consensus sequence of the sequenced stock was used. All positions along the genome where coverage reached at least 1,000 were considered. Mutations were identified where the aligned base was different from the ancestor. The frequency of the mutation was calculated as the number reads with the different base call divided by the total reads mapping to the position. Only mutations that reach at least a fraction of 0.01 were stored, although for most analyses, higher thresholds were used. Minority mutations were classed as mutations reaching at least 0.05, and majority mutations are mutations that reach at least 0.5.

Reversion positions were identified by aligning and comparing the consensus sequence from the HIV sequence database (HIV.lanl.gov, premade consensus/ancestral alignment of all subtypes, year 2004. Accessed June 18, 2019) gene per gene to the consensus of the sequenced virus stock. If, in later passages, this position mutated to the base in the consensus, then this is characterized as a reversion mutation.

## Simulations of neutral sequence evolution

Simulations of neutral sequence evolution were performed using the simulation tool described in [44]. In brief, a simulation is started with 100 virus particles with identical sequences (in this case, the consensus sequence of the ancestral virus). In every generation, every virus particle generates 10 offspring. Every offspring virion randomly acquires mutations according to the mutation rate. At every generation, the population is reduced to 100 to simulate the transfer in the experiment.

In total, 32 simulations were performed to match the data, 4 of 300 passages, 8 of 270 passages, and 4 of 240 passages, in each case using 2 mutation rates $10^{-5}$ and $10^{-4}$.

## Randomization tests

**Contribution of environment.** For the contribution of environment, majority mutations (mutations occurring at a frequency over 0.5 in any of the lines considered) were classified. All evolutionary lines were considered, but mutations shared between MT-2.A/MT-2.B or MT-4. A/MT-4.B only were considered not shared due to the confounding effect of a likely contamination event (see S1 Text). For the derived lines that changed environments, only new mutations after the switch were counted, i.e., mutations that reach over 1% before the switch are not counted.

If a mutation occurs at a frequency of over 0.5 in more than one of the evolutionary independent lines considered, it is classified as shared. Otherwise, it is classified as unique. Additionally, it is classified according to the environments (MT-2 and/orMT-4) in which this mutation is observed as a majority mutation.

Randomization tests were used to analyze whether there are more mutations in any of the classes than would be expected if environment had no effect. For this, the lines in which each mutation occurs were randomized. The number of lines each mutation occurs in was kept, but the lines were reassigned weighted by the frequency of mutations in each line. This means that the amount of majority mutation per line remains approximately constant, but the exact mutations are randomized—removing the potential effect of environment. Mutations were then classified as before. The dataset was randomized and classified in this manner 1,000 times, after which significance values were calculated from where in the distribution of randomized values the real value lies (2-sided test, if less than 2.5% of the randomized values lie above or below the real value, it is significant at the 0.05 level)

**Always- and never-combinations.** Always-combinations were calculated from mutations occurring in replicate A, 1 and 2 in MT-2 and MT-4, and this to prevent false signal from lines

sharing some history, either due to the switch or a contamination. For all mutations that reach majority in more than 1 line (the focus mutation), we identify the background. This is the set of all mutations that have a higher fraction at 2 time points: When the focus mutation reaches a fraction of 0.05 and when the focus mutation reaches a fraction of 0.5. These thresholds were somewhat arbitrarily chosen to balance calculability and accuracy. Always mutations are then mutations that are present in the background in all lines where this mutation reaches majority. For the randomization, we considered all mutations that reach these thresholds by the same time across all lines considered. Per line where the focus mutation reaches majority, we randomly draw a background from these mutations, counting how many are present everywhere.

For the never-combinations, we considered all lines, since shared history will not affect these measures. A never-combination is a mutation $c$ for which $f_{ct} < 1 - f_{ft} \; \forall \; t$, where $f_{ct}$ is the fraction of mutation $c$ at passage $t$ and $f_{ft}$ is the fraction of the focus mutation at passage $t$. This ensures that the 2 mutations never reach a frequency in the same line at which they must occur together. For the randomization, we again considered all mutations that are below this threshold across all lines considered, drawing the same number per line and redoing the analysis.

For both always- and never-combinations, $p$-values were calculated using the Wilcoxon signed-rank test (from the SciPy package ([www.scipy.org](www.scipy.org)) in Python) on the differences between data and randomizations.

## Supporting information

**S1 Text. Description and analysis of a contamination event in experiment I of the HIV long-term evolution experiment.**
(PDF)

**S2 Text. Analysis of the effects of the innate immune proteins APOBEC and ZAP in the HIV long-term evolution experiment.**
(PDF)

**S3 Text. Analysis of growth rates in the HIV long-term evolution experiment.**
(PDF)

## Author Contributions

**Conceptualization:** Karin J. Metzner, Roland R. Regoes.

**Data curation:** Eva Bons.

**Formal analysis:** Eva Bons.

**Funding acquisition:** Roland R. Regoes.

**Investigation:** Eva Bons, Christine Leemann.

**Methodology:** Karin J. Metzner, Roland R. Regoes.

**Resources:** Karin J. Metzner.

**Software:** Eva Bons.

**Supervision:** Karin J. Metzner, Roland R. Regoes.

**Visualization:** Eva Bons.

**Writing – original draft:** Eva Bons.

**Writing – review & editing:** Eva Bons, Karin J. Metzner, Roland R. Regoes.

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
