## [Editor Report · Decision Letter 0]

1 May 2020

Dear Dr Bons, 

Thank you for submitting your manuscript entitled "The Importance of Environment and Mutational History on Evolution of HIV-1 in a Long-Term Experiment" for consideration as a Research Article by PLOS Biology.

Your manuscript has now been evaluated by the PLOS Biology editorial staff, as well as by an academic editor with relevant expertise, and I'm writing to let you know that we would like to send your submission out for external peer review.

Please re-submit your manuscript within two working days, i.e. by May 05 2020 11:59PM.

Kind regards,

Roli Roberts

Senior Editor

PLOS Biology

---

## [Decision Letter · Decision Letter 1]

8 Jul 2020

Dear Dr Bons,

Thank you very much for submitting your manuscript "The Importance of Environment and Mutational History on Evolution of HIV-1 in a Long-Term Experiment" for consideration as a Research Article at PLOS Biology. Your manuscript has been evaluated by the PLOS Biology editors, an Academic Editor with relevant expertise, and by three independent reviewers.

You'll see that all three reviewers see value in your work, but seem to be struggling to identify what exactly we learn from the long-term nature of your study; in revising your manuscript, you should aim to address this primary shared concern, plus the other issues raised by the reviewers. In addition to the reviewers' comments, the Academic Editor has asked me to convey the following points:

a) Please could you make quantitative predictions of the number of fixed and segregating alleles in your populations, assuming a neutral (mutation/drift) model, and compare these predictions against the observed numbers of mutations? (this is an extension of a request made on a previous version of the manuscript that was not reviewed)

b) Rather than performing competition assays, as suggested by reviewer #3, rewording should suffice (e.g. "growth rate" instead of "fitness").

In light of the reviews (below), we will not be able to accept the current version of the manuscript, but we would welcome re-submission of a much-revised version that takes into account the reviewers' comments. We cannot make any decision about publication until we have seen the revised manuscript and your response to the reviewers' comments. Your revised manuscript is also likely to be sent for further evaluation by the reviewers.

We expect to receive your revised manuscript within 2 months. 

**IMPORTANT - SUBMITTING YOUR REVISION**

*Re-submission Checklist*

*Published Peer Review*

*PLOS Data Policy*

*Blot and Gel Data Policy*

Sincerely,

Roli Roberts

Senior Editor

PLOS Biology

REVIEWERS' COMMENTS:

Reviewer #1:

In this second version of the manuscript the authors address the concerns raised by the journal editors, first, whether drift or selection was the main driver of evolution is not yet convincing, with detailed selection analyses and direct fitness assays needed, second, the reasons why diversity and evolutionary rate were cell type-dependent remain obscure.

The main conclusion of this study is that the HIV-1 fitness landscape is mostly smooth, with many paths towards the same fitness peak, that is, the same adaptive mutations are fixed in independent evolutionary lines. This finding is not surprising since virus propagation in cell culture is not neutral, viruses are subjected to the selective forces exerted by the cell. As showed HIV-1 drug selection there is clear convergent evolution in selecting resistance, similarly, virus coreceptor switch from an R5 to X4 tropism that occurs in most infected patients through infection and disease progression is also an experiment of convergent phenotypic and genetic evolution, moreover, resistance selection and coreceptor switch are observed in the same time scale in most infected patients. Regarding the reasons why diversity and evolutionary rate were cell type-dependent is not also surprising since evolutionary forces may be different between cell types and the virus may also respond differently, however, these two cell types also share similar evolutionary forces (e.g. innate response, restriction factors, etc) and therefore, as showed here, at the end of the experiment viruses passaged in different cell types share most of the observed mutations. Say that, although with some concerns (see below), this reviewer believes that this is an interesting paper that may seed further evolutionary studies and, therefore, merits its publication on Plos Biology.

Specific comments

1. As a virologist it is unclear for me whether the virus fitness differences displayed in figure S5 are statistically significant. The authors should calculate the slopes of the virus kinetic experiments and confront them in a statistical test. To this end, replicas of the kinetic experiments should be performed.

2. Why are synonymous mutations shared in different experiments? (last paragraph on page 5) Are they adaptive? The authors may discuss this finding, are they affecting virus fitness?

3. MT4 and MT2 cells express ZAP, is pNL4-3 getting rid of CpG pairs through passages? Authors may discuss on virus nucleotide and dinucleotide frequencies.

4. Finally, I am guessing whether it would be of interest to employ not only two different cell types but also two different virus genetic backgrounds.

5. There are some in silico methods, simulations of neutral sequence evolution and randomization tests, that I can not evaluate because I am not familiar with them.

Reviewer #2:

Major comments:

1. I have some concerns about the extent of new ground broken by the study. The main findings (high degree of parallelism in evolved virus lines, virus evolution shaped by environment and historical contingency) are similar to observations from previous experimental evolution studies with RNA viruses (e.g. Sanjuan et al. 2004; Sanjuan et al. 2005; da Silva et al. 2010; Morley and Turner 2017). The long-term nature of this experiment does make it very unique, but I felt an opportunity was missed to foreground what this long-term experiment can teach us relative to typical shorter evolution experiments (see Wiser et al. 2013; Tenaillon et al. 2016 for examples of these long-term patterns from Lenski's LTEE).

2. This article included descriptions of temporal patterns of molecular evolution (section 2.3), but was lacking in quantitative and statistical comparisons. For example, there was no statistical comparison of the curves in Figure 2 to support the observations of differences between treatments. The dynamics of selective sweeps in Figure 3 could also be described quantitatively (see Lang et al. 2013; Levy et al. 2015; Acevedo et al 2014; McDonald et al. 2016).

4. The main question in this study is how environment and historical contingency interact to shape evolutionary trajectories. Differences in evolutionary history are built into the experiment by incorporating derived virus lines that are switched into different environments. However, with the exception of Figure 2, lines with different histories were not separated out during analysis. This makes the effect of evolutionary history less clear.

Minor comments:

1. In the first paragraph of the introduction, the authors state "Due to its high mutation rate, small genome, and large population sizes, it can quickly adapt to these changing selection pressures." It is not clear to me that a small genome results in quicker adaptation. 

2. "A high amount of environmentally robust mutations will lead to high levels of parallel or convergent evolution…" Clarification is needed here. Is the 'high amount of robust mutations' referring to the fraction of all possible mutations that are environmentally robust?

3. It would be helpful if figures were numbered in the order in which they are cited in the text. For example, Fig 6 is the second figure cited in the text.

4. It was unclear to me why fitness measurements were only reported for a subset of the experimental lines. For example, it would be interesting to see fitness data for the virus lines that experienced an environmental switch. 

5. In Section 2.2, the definition of 'reversions' requires clarification. Reversion mutations are defined in the text as 'mutations back to the consensus at positions where the ancestral virus differs from the consensus HIV sequence.' Do the authors mean mutations back to the ancestral sequence?

6. The plateau in the left panel of Figure 2 is very interesting. What do the authors think is going on there? This is an intriguing pattern that merits further discussion. 

7. In the legend for Figure 2, please clarify whether mutations included in the accumulation curves are relative to the ancestral sequence or consensus sequence at a given time point.

Reviewer #3: 

The main claims of this paper, in brief, are that HIV long-term adaptation in different cell environments appears to be non-neutral, environment specific, and that a considerable degree of parallel evolution occurs in small environmental changes. Ultimately, the claims are not novel because each of these points have been made in many other virus/host environment systems; where this work differs is that the passaging series is much longer than most studies, spanning hundreds of passages.

For a broader audience, this paper is almost exclusively focused on HIV, but seems to want to address broad topics of virus evolution (environment-specific adaptation, role of genetic background in accumulation of mutations, parallel evolution). A few references are made, but there is a very large body of work that covers this in many other viruses (and even other microbes). The authors could rewrite some parts to place their work in this broader context and make the paper more interesting to others.

Some other points to consider:

The two host environments are very similar environments. They are the same type of cell from two different origins. Most other work in virus evolution in different host environments uses different cell types, or even different host organisms - the difference between two clones of one cell type would not be considered 'significantly' different by many virologists, so I suggest providing information on how these two environments really differ from each other if it is known, or to include a comment on how little they differ in the discussion.

Parallel evolution in this case would be expected.

The fitness measures of fig S5 are not relative fitness measures, no competition experiments are performed, they are only replication kinetic experiments. It is known that viruses with small, yet significant, fitness differences have the same replication kinetics when they are grown alone; but a direct competition assay would reveal these differences. This test could be possible with a genetically marked NL4.3 clone. I leave it up to the authors and editor to determine whether this is required. In absence, I would think it safer to refer to the work as measures of replication, and not measures of fitness.

Finally, it is important to emphasize that the study uses only a single, lab adapted strain (NL4.3). All of the observations and conclusions should be viewed with this in mind, and this should be clearly addressed in the discussion. NL4.3 is not a representative of most HIV strains carried by individuals, and is already well adapted to growth in cell culture conditions.

Two points I'd like to mention that I found quite interesting:

-The authors show that APOBEC-induced mutations are not relevant among mutations that reach majority.

-In section 2.3, the data nicely show that when one passage series is switched to the other cell type, the appearance of >5% minority mutations is specific to cell type. These results are very clear and really shows how even a small difference in environment can have a primary influence on a virus's evolution.

---

## [Editor Report · Decision Letter 2]

27 Oct 2020

Dear Dr Bons,

Thank you for submitting your revised Research Article entitled "The Importance of Environment and Mutational History on Evolution of HIV-1 in a Long-Term Experiment" for publication in PLOS Biology. I have now discussed your revisions with the Academic Editor. 

On the basis of the Academic Editor's assessment, we're delighted to let you know that we're now editorially satisfied with your manuscript. However before we can formally accept your paper and consider it "in press", we also need to ensure that your article conforms to our guidelines. A member of our team will be in touch shortly with a set of requests. As we can't proceed until these requirements are met, your swift response will help prevent delays to publication. Please also make sure to address the data and other policy-related requests noted at the end of this email.

IMPORTANT: Please can you attended to the following:

a) Please can you make your title more declarative and appealing, e.g. "Long-term experimental evolution of HIV reveals... X, Y, Z"

b) Please attend to my Data Policy requests at the foot of this email.

c) Your Zenodo deposition is not currently accessible. Please could you send us a reviewer link or password so that we can check it?

- a cover letter that should detail your responses to any editorial requests, if applicable

*Copyediting*

*Published Peer Review History*

*Early Version*

Sincerely,

Roli Roberts

Senior Editor,

rroberts@plos.org,

PLOS Biology

DATA POLICY:

Regardless of the method selected, please ensure that you provide the individual numerical values that underlie the summary data displayed in the following figure panels as they are essential for readers to assess your analysis and to reproduce it: Figs 2-6, S1-S7. NOTE: the numerical data provided should include all replicates AND the way in which the plotted mean and errors were derived (it should not present only the mean/average values).

---

## [Editor Report · Decision Letter 3]

30 Nov 2020

Dear Dr Bons,

On behalf of my colleagues and the Academic Editor, Rafael Sanjuán, I am pleased to inform you that we will be delighted to publish your Research Article in PLOS Biology. 

PRODUCTION PROCESS

Before publication you will see the copyedited word document (within 5 business days) and a PDF proof shortly after that. The copyeditor will be in touch shortly before sending you the copyedited Word document. We will make some revisions at copyediting stage to conform to our general style, and for clarification. When you receive this version you should check and revise it very carefully, including figures, tables, references, and supporting information, because corrections at the next stage (proofs) will be strictly limited to (1) errors in author names or affiliations, (2) errors of scientific fact that would cause misunderstandings to readers, and (3) printer's (introduced) errors. Please return the copyedited file within 2 business days in order to ensure timely delivery of the PDF proof. 

If you are likely to be away when either this document or the proof is sent, please ensure we have contact information of a second person, as we will need you to respond quickly at each point. Given the disruptions resulting from the ongoing COVID-19 pandemic, there may be delays in the production process. We apologise in advance for any inconvenience caused and will do our best to minimize impact as far as possible.

EARLY VERSION

PRESS 

Kind regards,

Erin O'Loughlin

Publishing Editor, 

PLOS Biology

on behalf of

Roland Roberts,

Senior Editor

PLOS Biology